# The Link Between the Applied Visual Strategy When Copying the Rey–Osterrieth Complex Figure and the Language Abilities in Children with Specific Language Impairment

**DOI:** 10.3390/diagnostics15070851

**Published:** 2025-03-27

**Authors:** Ivana Milanović, Milena Paštar, Saška Žunić, Maša Marisavljević, Mile Vuković, Vladimir Janjić, Milan Đorđić, Miško Subotić

**Affiliations:** 1Cognitive Neuroscience Department, Research and Development Institute “Life Activities Advancement Institute”, 11000 Belgrade, Serbia; m.pastar@add-for-life.com (M.P.); s.fatic@add-for-life.com (S.Ž.); m.marisavljevic@add-for-life.com (M.M.); m.subotic@add-for-life.com (M.S.); 2Department of Speech, Language and Hearing Sciences, Institute for Experimental Phonetics and Speech Pathology “Đorđe Kostić”, 11000 Belgrade, Serbia; 3Faculty of Special Education and Rehabilitation, University of Belgrade, 11000 Belgrade, Serbia; mvukovic.dr@gmail.com; 4Department of Communication Skills, Ethics and Psychology, Faculty of Medical Sciences, University of Kragujevac, 34000 Kragujevac, Serbia; vladadok@yahoo.com (V.J.); mcpikac@yahoo.com (M.Đ.); 5Department of Psychiatry, Faculty of Medical Sciences, University of Kragujevac, 34000 Kragujevac, Serbia

**Keywords:** specific language impairment, visual perception, visual organization, language abilities, Rey–Osterrieth complex figure

## Abstract

**Background/Objectives**: Although specific language impairment (SLI) was thought to be a language impairment, recent studies suggest that it is also associated with domain-general and nonverbal deficits such as deficits in nonverbal working memory, visual short-term memory, executive functions, etc. This study aimed to examine if applied visual strategy when copying the Rey–Osterrieth complex figure (ROCF) correlates with language abilities in children with SLI. **Methods**: The sample consisted of 37 children diagnosed with SLI, divided into two groups based on the strategy used when copying ROCF. We used ROCF to assess perceptual organization and planning, and the Peabody Picture Vocabulary Test, Boston Naming Test, Token Test, Grammatical Judgment, The Children’s Grammar, and Global Articulation Test for language measurement. Univariate ANOVA was used for statistical analysis. **Results**: The results indicate that children who used a more mature strategy when copying ROCF achieved better results on tests used to assess grammar and articulation status. **Conclusions**: These results support the conclusion that there are neurocognitive mechanisms underlying both grammatical and visuospatial deficits. The obtained results suggest the importance of examining visual and visuospatial functions in children with SLI and the need for more comprehensive treatment of those children.

## 1. Introduction

Specific language impairment (SLI) is a neurodevelopmental disorder that affects understanding and/or producing language [1,2], which is not due to intellectual disability, neurological impairments, hearing impairments [2,3,4], social deprivation, or impairments in reciprocal social interactions [5]. This condition manifests as deficits in syntax, morphology, semantics, and phonology [6]. Comprehension deficits manifest as deficits in comprehending syntactically complex questions [7], questions with greater length [8], and object questions [9,10]; in comprehending passive sentences [11]; and in comprehending structures with noncanonical word order [8,11].

The expressive language of children with SLI is characterized by short, agrammatical, and incomplete sentences [12]. Studies have shown that agrammatism in children with SLI is manifested with deficits in noun-verb agreement [13], plural markers [13], noun cases [14], and the absence or inadequate use of function words [14]. In addition to difficulties in constructing grammatical structures, children with SLI also struggle with grammatical judgment [13,15].

Studies investigating the lexical abilities of children with SLI have shown that they experience challenges in lexical acquisition [16,17], difficulties in acquiring word meanings [16,18], and word definitions [16,19,20]. Moreover, they exhibit significantly smaller vocabulary sizes than typically developing children [16,17,21] and the reduced use of nouns and verbs [22].

In terms of the phonetic-phonological level, children with SLI exhibit deficits in phonological perception [4]. They also show abnormalities in speech production such as mispronunciations, omissions, substitutions, or inversions of phonemes, significantly impairing others’ ability to understand the speech of these children [21].

Although the term SLI implies that the essence of SLI is a language disorder, a growing body of studies has investigated the presence of non-linguistic factors [15,23,24,25,26,27], including auditory abilities [28,29,30], visual abilities [13,24,31], cognitive functions [1,3,13,24,25,27,32], and motor abilities [17,33]. The findings of these studies suggest the necessity of a different approach [25] concerning the definition, diagnostic criteria, and classification [27]. However, due to the large variety of research into this area, the current paper considers further only the research on visual abilities in SLI since it is the most relevant to the present investigation.

Studies have shown that children with SLI exhibit deficits in the visuospatial component of working memory [26,34,35]. Conducting a meta-analysis, Vugs et al. [35] indicated that most studies demonstrate that children with SLI perform worse on visuospatial working memory tasks than their typically developing peers. Nickisch and von Kries [31] found that children with SLI have impaired auditory and visual short-term memory. Contrary to previously mentioned studies, some authors proposed that impairment of working memory subcomponents is restricted only to damage to the verbal central executive system, as children with SLI performed similarly to their typically developing peers on visuospatial tasks [32,36].

Studies investigating the relationship between visual abilities and language abilities have presented various results. When investigating the correlation between visual short-term memory and language abilities, Nickisch and von Kries [31] found that visual short-term memory is one of the predictors of receptive language. These results suggest that visual short-term memory and auditory memory have independent impacts on receptive language. An alternative explanation could be the existence of a third factor necessary for understanding, which is also directly associated with auditory and visual short-term memory.

Similar to these results are results from a study by Fahiem and Mohammed [27], who emphasized the association between SLI and non-verbal abilities. These authors found that the semantic-pragmatic and syntactic phonological types of SLI showed a significant deficit among visuospatial abilities and spatial working memory. Vugs et al. [37] examined the correlation between working memory components and language abilities and found a low correlation between visuospatial storage and expressive vocabulary. Both verbal and visuospatial working memory deficits suggest a deficit in overall processing capacity, manifesting whenever tasks exceed processing capacities [38,39].

By examining the ability of children with SLI to recognize the grammaticality/ungrammaticality of sentences, Weismer et al. [13] found that nonverbal working memory predicts morphosyntactic abilities regarding error sensitivity and reaction speed. The authors of this study suggest that individual differences in nonverbal working memory predict how accurately and/or quickly children detect morphosyntactic errors. Schaeffer [40] also highlights the association between nonverbal working memory and grammar in children with SLI.

One of the tests that is frequently used to assess visual abilities is the Rey–Osterrieth Complex Figure Test (ROCF). This neuropsychological test is designed to evaluate visual memory and visuospatial constructive abilities. Also, it provides significant qualitative information about the strategies employed during the copying and recalling of complex figures and organizational approaches since it consists of multiple components that can be perceived as gestalt or separate details. Hence, executive functions, particularly planning and organization, are also examined through this test [41]. Since visual abilities have a significant role in speech development [42], further investigation of visual abilities in children with SLI is necessary. Therefore, ROCF can be a valuable tool for assessing visual abilities in children with SLI.

When using the ROCF and investigating a correlation between grammar abilities and visuospatial skills, Kiselev [43] found a correlation between grammar comprehension and visuospatial abilities. In another study, Kiselev and Glozman [15] utilized the ROCF for assessing holistic abilities, alongside “Comprehension of grammatical structures” from Luria’s neuropsychological assessment batteries. The results of their study indicated that children with grammar deficits use immature strategies when copying the ROCF. Given these results, the authors concluded that weakness in holistic synthesis can explain the problems in different abilities, including visuospatial abilities and grammar understanding [15]. On the other hand, authors who investigated visual abilities in children with ADHD [44] also noted the use of immature strategies in these children, which is expected for younger children. Additionally, children with ADHD reproduced the parts of the figure without organizing them as a whole. These authors explain this with the process of encoding information that can be affected by the executive deficit, which is typical of this diagnosis [44]. According to Akshoomoff et al. [24], using immature coping strategies in the ROCF suggests inefficient visuospatial processing, which may arise secondarily due to attention and planning deficits.

Considering the variety of results regarding visual abilities and their relationship with language abilities in children with SLI, it is necessary to expand research from linguistic abilities to non-linguistic functions and determine to what extent non-linguistic factors may influence the development of linguistic abilities [24]. Furthermore, it is still uncertain whether the same neurocognitive mechanisms underlie linguistic and non-linguistic abilities or whether these abilities are merely correlated. As some authors [42] suggest, human speech is a multisensory experience, and the most important modalities for language comprehension and production are visual spatial modalities. Therefore, studying visual abilities may be crucial for a better understanding of linguistic abilities. We hope to provide insights into how visual processing may support language abilities and whether these processes share common neurocognitive mechanisms. This study aimed to determine whether the strategy employed during copying the Rey–Osterrieth Complex Figure (ROCF) correlates, and to what extent, with linguistic abilities in children with SLI.

## 2. Materials and Methods

### 2.1. Study Design and Participants

The research initially involved 39 children with SLI who were receiving speech-language therapy at the Institute for Experimental Phonetics and Speech Pathology in Belgrade “Đorđe Kostić”, ages 6:1 to 8:11. All subjects included in the sample were diagnosed with SLI by a qualified specialist (speech-language therapist) by using tests to assess speech and language development. The study’s inclusion criteria required participants to be monolingual native Serbian speakers, with a diagnosis of SLI, average or above average IQ, and age over six years. Exclusion criteria were below-average intellectual functioning, hearing and vision impairment, brain damage or other neurological diseases, the presence of motor disorders, and genetic anomalies. To minimize the potential impact of confounders, we reduced their influence by ensuring that all participants were matched in terms of socioeconomic status, treatment duration, and handedness. All participants were treated in our clinic for approximately one year, and none had received previous speech-language therapy before being admitted to our institution. Furthermore, according to the Edinburgh Inventory [45], all participants were right-handed. Subjects were divided into two groups based on their strategy when copying the ROCF. Among the respondents, 3 respondents (7.1%) used strategy I, 18 respondents (42.9%) used strategy II, 19 respondents (45.2%) used strategy IV, and 2 respondents (4.8%) used strategy VII. Strategies II and IV were used by 88.1%, while the remaining strategies (I and VII strategies) were used by only 11.9% of participants. Therefore, strategy I and VII were excluded from further statistical analysis. Therefore, the final number of subjects was 37 children with SLI who were divided into two groups based on the strategy used on the ROCF: the first group consisted of children who copied ROCF using strategy II (first copying the details that are related to central rectangle or with some part of it, then finishing the rectangle and adding parts). The second group consisted of children who used the IV strategy while copying the ROCF (arranging the details in the drawing without an organized structure). The average age of the respondents was 84.62 months (SD = 12.24).

### 2.2. Measurements

#### 2.2.1. Visuoperceptual Skills Assessment

Rey–Osterrieth Complex Figure—ROCF [46] was used to assess perceptual organization and planning. The subject is given a white piece of paper and ROCF, which does not resemble any known object, and 5–6 colored felt-tip pens. The participant is instructed to copy a horizontally placed figure as precisely as possible and to reproduce it from memory after a certain time. For this research, only the first part was used, i.e., direct figure copying. The subject is given the task of copying the given figure while the examiner monitors the execution of the task. Each time the subject completes one part of the figure, the examiner switches the pen and notes the order in which the subject uses the pens. The total score is obtained by summing points for each successfully drawn detail of the figure. Eighteen details are scored with 0, ½, or 2 points according to their presence, accuracy, and placement. Also, the strategy used to copy the figure is noted. All strategies that can be used when copying ROCF are listed in Appendix A.

#### 2.2.2. Speech-Language Assessment

Speech and language were assessed using the tests used in clinical practice for decades in Serbia, though not all are standardized.

The Peabody Picture Vocabulary Test—PPVT-III-HR [47] is a standardized test that has been validated for use in children with SLI and is frequently employed as a tool for assessing receptive vocabulary, such as knowledge of nouns, verbs, and adjectives. Also, the test was validated for the Croatian-speaking population, and an adapted version is currently being used in Serbia. Since Serbian and Croatian belong to the same language group, the Croatian version adaptation was primarily based on certain term adjustments. The test has 204 tasks grouped into 17 strings with 12 words each. Black and white drawings are placed in front of the subject, i.e., four pictures, from which the subject must point out the picture representing a named object, action, or phenomenon. Examining is carried out until the subject makes eight mistakes within one string. The examination stops when he makes eight mistakes in one string for the first time, and a total score is formed. The raw score is obtained by summing the correct answers, while the standard score is obtained by converting the raw score, based on achievement and age.

The Boston Naming Test—BNT [48] was used to assess expressive vocabulary, i.e., object naming, in children and adults, with or without developmental and acquired speech and language impairments [49]. Although it has not been specifically validated for use with children with SLI, Isoaho, Kauppila [50], and Vukovic, Vukovic [51] used it in assessing naming abilities in children with SLI. The test consists of 60 black-and-white drawings of objects and assesses confrontational naming ability. Drawings are placed in front of the subject, who is asked to name them. Suppose the subject cannot name the object from the drawing. In that case, he can be given semantic support (explaining the term through its description/function, for example, the explanation “What do we sleep on” is used for the bed) or phonological support (given initial phoneme). All correct answers given by the subjects without support, with semantic support, and with phonological support are scored. Only the answers the respondents gave without support were scored, for this research. By summing up the correct answers, a total score was obtained.

A shortened version of the Token Test [52] was used to assess auditory comprehension. It is not officially standardized or validated for the Serbian language. However, it is used in clinical practice in Serbia to assess auditory comprehension in children with SLI [8,10]. The test consists of 39 verbal commands, which are divided into six subscales, ranked according to the degree of morphosyntactic complexity of the commands. Plastic tokens of different sizes (small and large), colors (white, red, green, blue, yellow), and shapes (circle and rectangle) are placed in front of the subject. The subject is then given verbal commands that require the subject to perform a specific activity with the tokens (e.g., show them, touch them, pick them up, touch each other, put them on top of each other…). The tokens are placed in a fixed order according to the pre-existing rules. In 4 subscales, all 20 tokens are used, while in the other four subscales, only 10 tokens are used, i.e., only large tokens. Before the examination, it is determined whether the subject distinguishes the features of the visual material, i.e., whether he distinguishes shapes and colors. When it is determined that it differs, the respondent is given an example to understand how the order is executed. After the subject manipulates the tokens, the examiner returns them to their initial position. Achievement on each task is graded as correct or incorrect. By summing up the completed tasks, the total score is obtained.

Grammatical Judgment is a test constructed by the author for research purposes. The test consists of 40 sentences, of which 20 are grammatical, and 20 are ungrammatical. The sentences are formulated to be concise, not burdening the respondents’ working memory. Agrammatic sentences are composed in such a way that the order of the words in the sentence is not by the grammar of the Serbian language, or they contain inadequate use of functional words (he came from his mother/the owl lives in the forest/we will go the cinema) or inadequate morphological endings for the form. The subject reads a sentence and is asked to say whether the sentence is correct or not. Before testing, the test taker is given an example sentence to determine if they have understood the task. The respondent is given the example “The mouse eats ham_” (with omission of the morphological mark for case) and is asked if it is adequately said. After the respondent says the sentence adequately, the testing starts. Each sentence is scored as pass or fail. The total score is obtained by summing up the successful answers. The test is given in Appendix B.

The Children’s Grammar [10,53,54] was used to assess the expressive component of grammar. This test was initially developed in Serbian and adapted for Serbian speakers. It is commonly used to assess grammatical development in various language disorders. The mentioned instrument evaluates the use of the plural, gender, case, verbal nouns, pronouns, adjectives, verb tenses, prepositions and adverbs, complex statements, interrogative sentences, and interrogative-negative sentences. Each of the mentioned tasks consists of visual material, where the examiner begins with pronouncing a particular sentence, which the child should complete by producing a particular grammatical form. If there is a need for it, the examiner provides the child with additional support in the form of an explanation of what is in the picture so that the child can adequately produce the required shape. All correct answers are scored; the score is obtained by adding up all successfully given answers. A detailed description of the instrument is given in the work of Bogavac et al. [54].

The Global Articulation Test—GAT [53,54,55] was used to assess articulatory status. This test was also initially developed in Serbia to assess the production of 30 sounds in the Serbian language. The test consists of 30 bisyllabic words where the tested sounds are positioned in medial (vowels) and initial (consonants) word positions. To identify the type and degree of pathological pronunciation, the examiner-speech therapist relies on auditory assessment of acoustic characteristics of the uttered sounds, simultaneously observing the position of the child’s speech organs during pronunciation. By summing correctly pronounced sounds, a score is obtained. A detailed description of the instrument is provided in the work by Rakonjac et al. [55].

#### 2.2.3. Cognitive Assessment

Cognitive skills were tested using Weschler’s Intelligence Scale for Children—Serbian version (REVISK; [56]). The verbal and nonverbal intelligence scores were assessed by an experienced child psychologist with experience in cognitive skills assessment.

### 2.3. Procedure

The research data were collected from February 2023 to September 2023 within the Institute for Experimental Phonetics and Pathology “Đorđe Kostić”. Subjects were tested in a quiet, plain room to eliminate most potential distractors. Only the participant and the examiner were present in the room. An experienced speech-language pathologist assessed visuo-perceptual and speech-language abilities, while an experienced psychologist assessed cognitive skills. Since six tests were applied, the testing was done over two days to not fatigue the participants. Before administering each test, the examiner provided examples to ensure that the subject understood what was being asked of him. When the examiner ensured that the subject understood the task, the research was started. If needed, the speech-language therapist additionally directed the child’s attention before giving the order or made additional breaks until the child was focused on the task again.

The complete study protocol was in accordance with the Ethical Principles in Medical Research Involving Human Subjects, as established by the Declaration of Helsinki, and had been approved by the Ethics Committee of the Institute for Experimental Phonetics and Speech Pathology in Belgrade, Serbia (No S-23-01). The children’s parents gave written informed consent for cognitive and speech-language assessment and participation in this study.

### 2.4. Statistical Analysis

All data were analyzed using the SPSS 20.0 software package [57]. The total score of all the applied tests was calculated. Then, the outliers were detected, and the scale’s reliability was assessed. After that, descriptive statistics were conducted, the normality of the distribution was tested, and accordingly, we made a decision to use the parametric Chi-square test two groups comparison with normal distribution in terms of gender, and Independent Samples *t*-test in terms of age and IQ. Differences between two groups in terms of investigated dependent variables were obtained using Univariate ANOVA. A *p*-value equal to or less than 0.05 was considered statistically significant.

## 3. Results

The results of the Independent Samples *t*-test showed that there are no statistically significant differences between groups regarding age (t (35) = 1.44, *p* = 0.16)), verbal IQ (t (35) = 1.64, *p* = 0.11) and nonverbal IQ (t (35) = 0.75, *p* = 0.46). The results of Chi-square test showed that there was no statistically significant difference between groups regarding gender (χ^2^ (1) = 0.51, rφ = 0.12, *p* = 0.48). The sample distribution regarding gender, age, and intellectual status (IQv and IQm) is given in Table 1.

By summing the correct answers, the results of each test were obtained. The average score on the ROCF for the entire sample was M = 15.88 (SD = 7.35), while that of group I was M = 19.78 (SD = 7.49), and that of group II was M = 12.18 (SD = 5.04). The *t*-test results for independent samples showed a statistically significant difference between children with RD who used different strategies regarding ROCF scores. Those who used strategy II (M = 19.78, SD = 7.49) showed statistically significantly better results (t (35) = 3.64, *p* = 0.001) than those who used strategy IV (M = 12.18, SD = 5.04). Average scores, measures of descriptive statistics for the ROCF, and applied speech-language tests are given in Table 2.

The results of the Univariate ANOVA showed that there is a statistically significant difference between the two groups of children who used a different strategy when copying the ROCF on the Grammatical Judgment test (F (1, 35) = 9.58, *p* = 0.004, η^2^ = 0.21), the Children’s Grammar (F (1, 35) = 9.70, *p* = 0.004, η^2^ = 0.22), and the GAT (F (1, 35) = 15.09, *p* < 0.001, η^2^ = 0.30). On other tests (PPVT-III-HR, BNT, and Token Test), no statistically significant difference was found between the two groups (*p* > 0.05).

## 4. Discussion

Our results showed that 11 children with SLI (45.8%) used a conceptual strategy, starting by tracing the rectangle and then adding associated details. Conversely, more children with SLI (54.2%) used a less mature strategy, namely a fragmented strategy characterized by drawing in fragments, part by part, related to the central rectangle. In other words, these children did not demonstrate an organized structure when copying the given figure. The fragmented strategy indicates a partial approach to the figure and an inability to integrate and connect parts into a whole [58]. The inability to integrate parts into a whole and perceive gestalt is associated with immature and undeveloped executive functions, particularly planning, conceptual reasoning, and problem-solving abilities [41,59]. Apart from the correlation between planning and using specific strategies, authors Larson, Gangopadhyay, Kaushanskaya, and Weismer [60] have found a link between planning abilities and language skills. They identified that children with SLI and poorer language abilities struggle more with planning [60]. Abdul Aziz and colleagues [61] have demonstrated that children with SLI follow a different and slower trajectory in developing inner speech, which is crucial for planning and directing behavior, serving as a verbal mediator [62].

It is traditionally believed that children with SLI exhibit a similar developmental pattern of executive functions as typically developing children, given the fact that SLI is exclusively characterized by a language development disorder [1]. However, a growing body of evidence suggests the presence of non-linguistic deficits in children with SLI [63,64]. For instance, Kiselev [43] suggests that children with SLI use less accurate, less mature, and fragmented strategies and exhibit subtle deficits in processing configurational information. Based on this, the authors suggest that children with grammatical SLI have a deficit in a specific brain mechanism responsible for holistic synthesis, resulting in simultaneous deficits in both linguistic and non-linguistic domains [15].

In the work of Akshoomoff et al. [24], poorer performance was observed in children with SLI on the ROCF compared to typically developing peers matched by age. These results indicate less mature and less efficient approaches to visuospatial tasks or subtle deficits in visuospatial processing tasks. Our study results showed that children using a more mature strategy also had more drawn elements or a higher score on this test, consistent with findings from other authors [65,66]. Some authors [25,26,37] confirm that children with SLI have deficits in executive functions, affecting both verbal and nonverbal components. Furthermore, children with SLI during visuospatial working memory tasks [28] exhibit impaired verbal working memory [3] and impaired visuospatial working memory due to reduced visuospatial storage and inefficient verbal coding.

When considering the impact of the applied copying strategy on language abilities, the results of our study showed a statistically significant difference between the groups examined in expressive and receptive components of grammar, as well as articulation status. In contrast, no statistically significant difference was found between groups regarding receptive and expressive vocabulary and comprehension. Children who used a more mature strategy achieved better results on all mentioned tests. The link between the chosen strategy and grammatical abilities finds its basis in Luria’s theory regarding the influence of spatial perception on grammatical abilities. According to Luria, these abilities may share underlying mechanisms. Both abilities require: 1. the ability to segment the input into parts; 2. understanding that certain parts can be assembled together as components of a recognizable structure of a known type; and 3. an understanding of the whole regarding the relationship between these parts [15]. According to the ROCF, the participant must perceive the parts and the whole of the presented figure. Similarly, in processing the grammar of spoken discourse, it is necessary to recognize each word and understand its relationship with other words to interpret adequately and understand the sentence. Understanding grammar and its correct usage requires simultaneous processing of words and morphemes received auditorily. Since children with SLI have difficulty in the simultaneous processing of auditorily presented information, it is considered that they cannot identify and understand constant changes in morphological and syntactic patterns. As a result, they struggle to follow all language rules along with the changes in the dynamic environment of conversation [6].

Furthermore, another potential explanation for such results is the inability to integrate audio-visual stimuli, which also contributes to language deficits in SLI. This explanation is consistent with the results of our study. Specifically, in the Grammatical Judgment and GAT, participants were required to understand and then successfully perform tasks based on visual inspection of the examiner and auditory presentation of stimuli. Grammatical Judgment was determining if a sentence was grammatically correct, and for GAT, it was repeating a word with a target phoneme while watching and listening to the examiner. Children with SLI who use less mature figure copying strategies may achieve lower scores on grammar and articulation tests because they cannot integrate what they hear and see, i.e., they cannot adequately process speech-language information. This difficulty is particularly observed when lip-reading, as they cannot process and respond to information appropriately. Similar to our results are findings from other authors. For instance, Norrix, Plante, Vance, and Boliek [28] examined the McGurk task in children with SLI. In this task, children were audibly presented with the syllable/bi/and visually presented with/gi/to determine whether participants relied more on auditory or visual information. Using the McGurk task, these authors found that children with SLI relied less on visual information than their typically developing peers. They concluded that, in addition to deficits in auditory perception exhibited by these children, they also experience difficulties in audio-visual processing. Furthermore, results from the same study indicate that children with SLI exhibit poorer performance than the control group in detecting audio-visual asynchronies in human and synthetic speech. Additionally, Kaganovich, Schumaker, and Rovland [67] also reported that children with SLI are less sensitive to matching auditory words with visual articulations. They suggest that this reflects difficulties in connecting what they hear (speech sounds) with what they see (visible movements of speech organs during articulation). Pons, Sanz-Torrent, Ferinu, Birules, and Andreu [68] conducted a study to investigate whether children with SLI might show reduced attention to the talker’s mouth. The results showed that these children did not demonstrate a preference for mouths or eyes. Their gaze was equally directed toward both, indicating a deficit in perceiving and integrating audio-visual cues in speech. In other words, these results indicate that children with SLI look less at mouths as a group than typically developing peers [68]. These findings are consistent with those of Heikkila et al. [69], which found that 7-year-old children with SLI look significantly less at lips compared to their typical peers. Similarly, Meronen et al. [70] reported reduced reliance on lip-reading in noisy situations.

Although participants in the first group who used a more mature strategy showed higher scores on each test, the observed differences on vocabulary tests (PPVT-III-HR and BNT) and comprehension tests (Token Test) were not statistically significantly better compared to the group of children with SLI who applied a less mature strategy. The nature of the tasks in the applied tests can explain these findings. The PPVT-III-HR test requires children to point out the requested term, i.e., “Show where the child is”, using visually presented stimuli (pictures with minimal detail). The BNT requires children to name pictures and answer a basic question, “What is this?” Children found it more accessible, and deficits in audio-visual integration were less noticeable when presented with tasks involving visually presented stimuli or pictures. During the Token Test, children looked at tokens placed in front of them and manipulated them. In contrast, tasks that required listening and observing the examiner (in tests assessing grammar and articulation status) more heavily challenged audio-visual integration. Therefore, it is possible that in tasks of that nature, visual strategies had a more significant impact on language abilities in grammar and articulation.

The results indicate the significance of visual perception and organization abilities in developing language skills, highlighting that these abilities cannot be viewed separately from linguistic functions. As noted by some authors, speech perception is a multimodal process that requires the integration of auditory and visual inputs [71]. Studies using the McGurk effect have also shown that audio-visual perception and integration are present even in very young children and play a significant role in developing speech and language abilities [72,73]. Audio-visual integration is also observed in older children and adults, and it plays an important role in understanding speech in everyday social situations. Although the study sample is small, the results can have practical implications and guide therapeutic work with children with SLI. Before starting the treatment, children should be assessed to determine their levels of both linguistic and non-linguistic abilities. Consequently, the treatment methodology should be adjusted to emphasize visual abilities and ensure that visual perception fulfils its function, ultimately improving the impact of visual perception in therapy. This will help integrate visual and auditory stimuli, improving speech perception. The treatment should also include exercises to improve the ability to perceive wholes. Furthermore, this study may provide direction for future research. Examining auditory perception alongside visual perception would contribute to a better understanding of the phenomenology of SLI and the specifics of auditory and visual stimulus processing. This approach could foster a holistic view of this disorder.

## 5. Conclusions

This study investigated the link between visual perception/organization and language abilities in children with SLI. Based on the results, the following conclusions can be drawn:Among children with SLI up to 9, two strategies were identified while copying the Rey–Osterrieth Complex Figure: some employed a mature strategy using gestalt principles to integrate parts into a whole. In contrast, others used a fragmented strategy, copying parts of the figure without clear organization or integration into a whole.Children with SLI who used a less mature drawing strategy particularly exhibited deficits in processing linguistic information requiring simultaneous listening and observing of the examiner.The results support the conclusion that there are neurocognitive mechanisms underlying both grammatical deficits and visuospatial deficits.

The results indicate a link between visual strategy and performance on speech-language tests based on audio-visual perception. However, it remains an open question whether visual perception is the cause of poorer outcomes or if both are rooted in shared neurocognitive processes. Further research is necessary to answer this question. Also, these findings suggest that children with SLI are not a homogeneous group in terms of visual strategies. They support the existence of perceptual and visuospatial deficits in SLI, not just linguistic deficits, as previously believed. Our study has identified perceptual, visuospatial, and audio-visual integration difficulties in individuals with SLI. However, the extent, nature, and significance of these difficulties in relation to SLI remain uncertain. It is unclear whether they represent core features of the condition or co-occurring characteristics influenced by other factors. Further research is needed to better understand their prevalence, underlying mechanisms, and potential relevance to classification and diagnosis. Furthermore, the findings of this study may have implications for speech-language intervention, suggesting that therapeutic approaches could benefit from integrating targeted support for visual-perceptual, visuospatial, and executive functions. Given the potential role of multisensory processing in language development, interventions that incorporate training in audio-visual integration may facilitate more efficient speech perception and linguistic processing. Additionally, structured activities designed to enhance visuospatial and perceptual skills, such as spatial reasoning tasks and visual tracking exercises, may contribute to improvements in cognitive processing. Moreover, interventions aimed at strengthening executive functions, including working memory, cognitive flexibility, and attentional control, could support broader language and cognitive outcomes. Adaptive therapeutic strategies that tailor interventions to individual cognitive profiles, incorporating structured visual cues and multimodal learning techniques, may further enhance intervention efficacy. However, the extent to which these approaches are effective in clinical practice remains to be determined, necessitating further empirical research to assess their impact and refine their implementation. These considerations may contribute to a more comprehensive framework for understanding and assessing SLI, although further empirical research is needed to validate their significance and clinical applicability.

## Figures and Tables

**Table 1 diagnostics-15-00851-t001:** Sample distribution regarding gender, age, and intellectual status (IQv and IQm).

	Gender		Age	IQm	IQv	Total
	m	f				
Group I	15 (83.3%)	3 (16.7%)	87.56 (12.38)	103.44 (15.24)	89.11 (12.06)	18
Group II	14 (73.7%)	5 (26.3%)	81.84 (11.75)	100.05 (12.15)	82.42 (12.72)	19

Note: The age of the respondents is shown in months.

**Table 2 diagnostics-15-00851-t002:** Descriptive statistics of all applied tests.

		M	SD	Min	Max
ROCF	Group I	19.78	7.49	6	33
	Group II	12.18	5.04	6	23.5
PPVT-III-HR	Group I	85.00	19.91	43	110
	Group II	85.16	12.39	64	104
BNT	Group I	27.22	6.25	14	36
	Group II	25.74	6.04	16	36
Token Test	Group I	95.67	34.54	28	135
	Group II	79.37	29.59	39	135
Grammatical Judgment	Group I	31.11	4.46	21	38
	Group II	25.32	6.65	11	37
The Children’s Grammar	Group I	18.94	3.06	13	23
	Group II	14.84	4.73	6	25
GAT	Group I	27.22	1.90	23	30
	Group II	24.58	2.44	20	30

## Data Availability

The data presented in this study are available at the request of the corresponding author for ethical reasons.

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
