# Peer review of "The Link Between the Applied Visual Strategy When Copying the Rey–Osterrieth Complex Figure and the Language Abilities in Children with Specific Language Impairment"

_diagnostics, 2025, doi:10.3390/diagnostics15070851_

Round 1
Reviewer 1 Report
Comments and Suggestions for Authors
The paper is devoted to the study of an important topic – the development of an additional method for diagnosing SLI based on the analysis of visual strategy of children when copying the Rey-Osterrieth complex figure.
I fully support the ideology of the study, aimed at finding complex disorders in SLI.
The authors present a detailed analysis of the literature on the topic of the study, including an overview of the methods used to analyze SLI, tests, and points of view on the problem.
The method is described in detail. The research included 26 children with SLI, ages 6-9 years. The criteria for inclusion of children in the study and their division into two groups based on their strategy when copying the ROCF are given. The reduction of the initial sample of children is justified. The tests and testing procedures used in the study are described. For visual - perceptual skills assessment, the Rey-Osterrieth Complex Figure test – ROCF was used, for speech-language assessment – different test: the Peabody picture vocabulary test - PPVT-III-HR, the Boston Naming Test – BNT, the shortened version of the Token Test, Grammatical Judgment, Children’s Grammar, and Global Articulation Test – GAT. Cognitive skills of children were tested by Serbian version of Weschler’s Intelligence Scale for Children.
The results of the study showed a statistically significant difference between the two groups of children who used a different strategy when copying the ROCF on the Grammatical Judgment test, the Children’s Grammar and the GAT. The authors consider conceptual strategy starting by tracing the rectangle first as more mature strategy vs fragmented strategy, based on the first copying of details. Children with SLI who used a fragmental strategy exhibit deficits in processing linguistic information. The results indicate a link between visual strategy and performance on speech-language tests based on audiovisual perception.
In the discussion, the authors compare the data obtained in the study with existing results on different samples of children.
In general, the results presented in the paper are important for the diagnosis of SLI. However, there are a number of remarks.
Major:
- The data on the control group (typically developing children) are absent.
- It would be necessary to test children with the 2nd and 3rd strategies, who were not included in the further study, since with an increase of the sample the number of children with 2nd and 3rd strategy could increase. These data are important for further work.
- As a discussion - was the children's functional lateral asymmetry profile checked - leading hand, eye, ear, and hemisphere? There are different strategies for processing information, and the organization of behavior depending on the leading hemisphere - right or left. If this is not possible, then data on the leading hand could be provided.
Minor:
The child was given 5-6 colored felt-tipped pens for the ROCF test. Is it important for the results’ estimation (was this taken into account in the analysis)?
Reviewer 2 Report
Comments and Suggestions for Authors
1) The introduction provides an extensive background on Specific Language Impairment (SLI) and its linguistic and non-linguistic deficits, but it does not clearly define what specific gap in the literature this study aims to address.
2) Clarification on ROCF Strategy Grouping: The study mentions that strategies I and VII were excluded due to small sample sizes. However, the justification for separating Strategies II and IV into two main groups should be further elaborated with theoretical or prior research support.
3) The introduction mentions ROCF but does not explicitly justify why this test is particularly relevant for studying SLI.
4) The introduction presents a long review of various studies, making it harder for readers to identify the study’s focus.
5) The study initially included 26 participants but excluded two due to rare strategy use. However, the final sample size of 24 is small, and there is no power analysis to justify whether this is sufficient for statistical reliability.
6) Language abilities were measured using multiple standardized tests, but there is no mention of whether the tests were adapted or validated for the specific population being studied (Serbian-speaking children with SLI).
7) The manuscript suggests that findings could help in therapy for children with SLI, but it does not clearly explain how.
8) The study reports statistical significance (p-values) but does not include effect sizes (e.g., Cohen’s d, eta squared). This makes it difficult to assess the practical significance of the findings.
9) The study does not address potential confounders, such as socio-economic background, bilingualism, or previous therapy experience, which could influence language development.
Some sentences are long and complex, making readability challenging, and Some sentences do not flow naturally and may be difficult to understand.
Round 2
Reviewer 2 Report
Comments and Suggestions for Authors
1) The discussion about revising SLI classification is interesting but should be framed more cautiously, as this study alone may not be enough to justify a diagnostic overhaul.
2) further elaboration on potential intervention strategies based on these findings would strengthen practical implications.
3) There are inconsistent uses of British and American English
4) Some sentences are long and complex, making it harder for the reader to follow. Consider breaking them into shorter, more digestible sentences.
